# Iron-based nanocatalyst for the acceptorless dehydrogenation reactions

Garima Jaiswal[1], Vinod G. Landge[1], Dinesh Jagadeesan[2] & Ekambaram Balaraman [1]

Development of sustainable catalytic systems for fundamentally important synthetic transformations and energy storage applications is an intellectually stimulating challenge. Catalytic dehydrogenation of feedstock chemicals, such as alcohols and amines to value-added products with the concomitant generation of dihydrogen is of much interest in the context of hydrogen economy and is an effective alternative to the classical oxidation reactions. Despite a number of homogeneous catalysts being identified for the acceptorless dehydrogenation, the use of high price and limited availability of precious metals and poor recovery of the catalyst have spurred interest in catalysis with more earth-abundant alternatives, especially iron. However, no report has described a reusable iron-based heterogeneous catalyst for oxidant-free and acceptorless dehydrogenation reactions. Here we replace expensive noble metal catalysts with an inexpensive, benign, and sustainable nanoscale iron catalyst for the efficient acceptorless dehydrogenation of N-heterocycles and alcohols with liberation of hydrogen gas.

[1] Catalysis Division, CSIR-National Chemical Laboratory (CSIR-NCL), Dr. Homi Bhabha Road, Pune 411008, India. [2] Physical and Materials Chemistry Division, CSIR-National Chemical Laboratory (CSIR-NCL), Dr. Homi Bhabha Road, Pune 411008, India. Correspondence and requests for materials should be addressed to D.J. (email: d.jagadeesan@ncl.res.in) or to E.B. (email: balaramane2002@yahoo.com)

Rapid depletion of limited fossil fuels has directed a significant amount of human efforts to identify alternative renewable, green energy resources. It has been widely accepted that hydrogen as a fuel can be effective to curtail the energy crisis if produced at the proximity of usage without involving long-distance transportation. Thus, the most attractive and efficient hydrogen storage/release media are liquid organic hydrogen carriers (LOHCs), because they have relatively high hydrogen content and can easily be transported. In this context, extraction of $H_2$ from abundant, renewable feedstocks via acceptorless dehydrogenation strategy (AD) is extremely important, but a thermodynamically uphill process[1–5]. Traditionally, extraction of hydrogen atoms in adjacent positions in an organic molecule can be achieved either by the use of a stoichiometric amount of strong oxidants or a sacrificial hydrogen acceptors, which often produce copious waste[6,7]. On the other hand, catalytic dehydrogenation with concomitant removal of dihydrogen is a superior strategy and has enabled the direct access to valuable intermediates.

Catalytic acceporless dehydrogenation by homogeneous transition–metal complexes has been a well-recognized sustainable process in chemistry[8–13]. Despite the satisfactory performance at the laboratory scale, homogeneous catalysts face severe limitations such as high cost, poor recovery, extensive synthesis of ligands, and poor stability while scaling up for industrial applications. In comparison, heterogeneous catalysts would be more advantageous with respect to catalyst recycling and reusability. Moreover, tremendous scope for improving step-economy, atom-economy, and benignity of AD strategy that could be more attractive to chemical industries and can be foreseen using the state-of-the-art catalysts. The economical, biorelevant, and ample supply of iron salts coupled with their lack of toxicity, makes them ideal candidates to replace precious metals for both the academic and industrial applications[14–18]. However, search for an efficient catalytic system for AD strategy using earth-abundant iron catalysts is exceedingly rare. The reason is that these have a propensity to undergo one electron redox process due to the narrow energy gap between the d-orbitals. This phenomenon may lead to undesired reaction pathways in the dehydrogenation reaction, and indeed, controlling the reactivity of the catalyst is very challenging. Many molecularly defined iron complexes with judiciously chosen ligands have been applied successfully under homogeneous conditions as an alternative to noble metal-based catalysts in recent times[16–18]. Unfortunately, most of these homogeneous complexes are rather sensitive and/or sophisticated, presenting synthetically challenging ligand systems (Fig. 1a, b).

Recently, Fe-based nanocatalyst supported on graphitic carbon has emerged as a valuable catalyst in hydrogenation and related reactions[19–23]. Early works in iron–graphite nanocatalyst systems focused on exploiting the high surface area and excellent transport properties of the graphene layers. Iron–carbon nanocatalyst system is highly sensitive to its microstructure and has been shown to significantly affect the catalytic outcome[23–25]. Thermal decomposition of a molecular complex of a metal on a support is a versatile method to obtain supported nanocatalyst. Based on this approach, $Fe_2O_3$- and Co-based catalysts encapsulated in N-doped graphitic carbon for hydrogenation and oxidation reactions are recently documented[21,22,26–39]. The final composition and the microstructure of the catalyst are observed to vary depending on the ligand, nature of carbon support, decomposition temperature, and reaction atmosphere. Here we show a unique core–shell architecture of iron nanocatalyst with a shell comprising of oxide and a core mainly of carbide synthesized by thermally pyrolyzing Fe:N-rich ligand on graphitic oxide support. Interestingly, the microstructure of the final catalyst showed a surface lacking the encapsulating sheath of carbon commonly observed in earlier works[21–25]. The unique microstructure also resulted in an exceptional catalytic property in oxidant-free and acceptorless dehydrogenation of N-heterocycles, relatively abundant alcohols, and amines with the concomitant generation of hydrogen gas. In addition to the concept of LOHC, this elegant system offers an alternative streamlined strategy for the sustainable production of chemicals with a great step-economy and reduced waste generation. To the best of our knowledge, the acceptorless dehydrogenation of diverse alcohols (1°, 2° alcohols and diols), amines, and N-heterocycles by a single iron-based catalyst either under homogeneous or heterogeneous conditions has not been investigated yet.

## Results

**Synthesis and characterization of the iron nanocatalyst**. The catalyst material was prepared by wet impregnation of 1:1 molar

**Fig. 1** Iron-catalyzed acceptorless dehydrogenation reactions. **a** Previous works describing homogeneous Fe complexes used for acceptorless dehydrogenation of alcohols. **b** Previous work involving Fe-catalyzed accceptorless dehydrogenation of N-heterocycles

**Table 1 Acceptorless dehydrogenation of 1,2,3,4-tetrahydroquinoline (1a)**

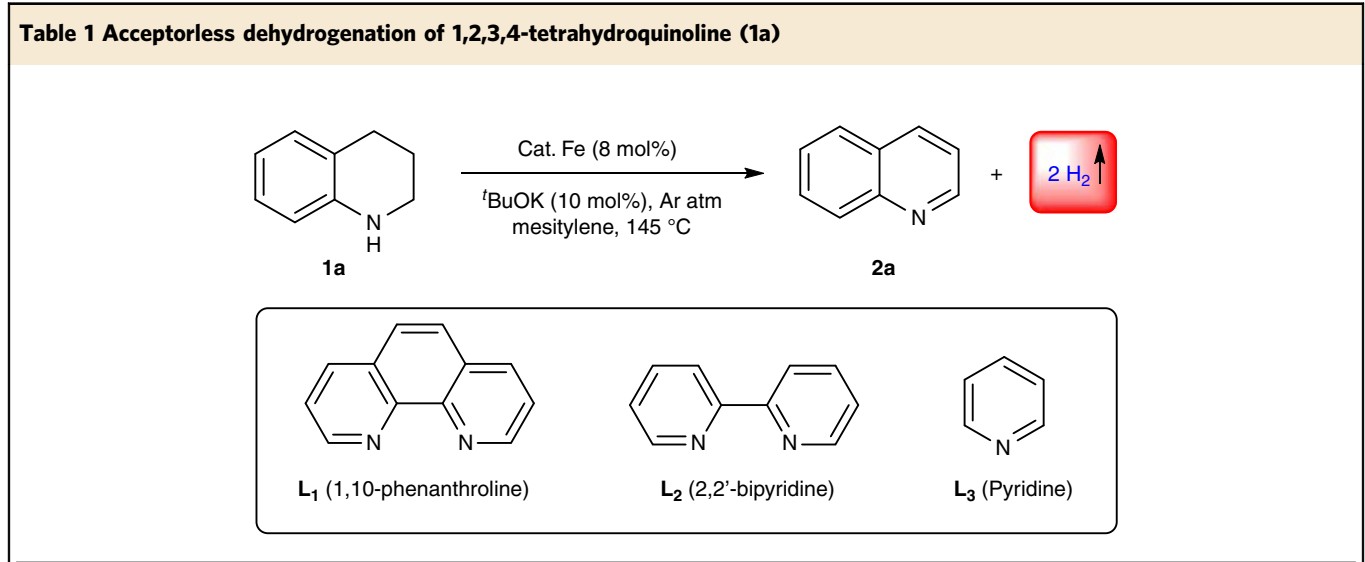

| Entry | Catalyst | Conversion (%)[a] | Yield (%)[a] |
|---|---|---|---|
| 1 | Fe-Phen[b] | Trace | Trace |
| 2 | Fe-Phen@EGO[c] | 20 | 16 |
| 3 | Fe@EGO | 17 | 15 |
| 4 | Phen@EGO | 6 | Trace |
| 5 | Fe-L$_1$@EGO-400 | 40 | 37 |
| 6 | Fe-L$_1$@EGO-600 | 52 | 45[d] |
| 7 | Fe-L$_1$@EGO-900 | 98 | 92 (88)[d] |
| 8 | Fe-L$_2$@EGO-900 | 51 | 40 |
| 9 | Fe-L$_3$@EGO-900 | 30 | 22 |
| 10 | — | 0 | 0 |
| 11 | Fe-L$_1$@Al$_2$O$_3$-900 | 23 | 19 |
| 12 | Fe-L$_1$@SiO$_2$-900 | 15 | 11 |
| 13 | Fe-L$_1$@CeO$_2$-900 | 8 | Trace |
| 14 | Fe-L$_1$@TiO$_2$-900 | 27 | 16 |

Reaction conditions: **1a** (0.5 mmol), cat. Fe-L$_1$@EGO-900 (8 mol%), t-BuOK (10 mol%), and mesitylene (2 mL) heated at 145 °C
[a]Yields of **2a** and conversion of **1a** were determined by gas chromatography (GC)
[b]Reaction under homogeneous conditions using the in situ-generated Fe catalyst
[c]Non-pyrolyzed materials
[d]Isolated yield

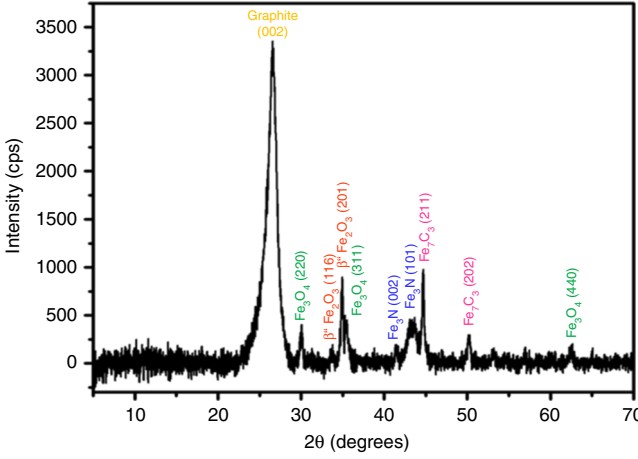

**Fig. 2** PXRD characterization. PXRD pattern of Fe-L$_1$@EGO-900 with indices of peaks with the pattern of Fe$_3$O$_4$, Fe$_3$N, β″-Fe$_2$O$_3$, Fe$_7$C$_3$, and graphite

combination of iron(III) acetylacetonate and 1,10-phenanthroline L$_1$ on exfoliated graphene oxide (EGO). Subsequent pyrolysis at 900 °C for 4 h under argon atmosphere formed nanoscale iron catalyst Fe-L$_1$@EGO-900 (Supplementary Fig. 105). Under

similar reaction condition, iron catalysts Fe-L$_2$@EGO-900 and Fe-L$_3$@EGO-900 were prepared using 2,2′-bipyridine (L$_2$), and pyridine (L$_3$) as ligands, respectively. Various parameters such as nature of the support, chemical composition, microstructure, and temperature of pyrolysis were also studied and correlated with their activity toward acceptorless dehydrogenation of N-heterocycles (Table 1).

The catalyst Fe-L$_1$@EGO-900 was extensively characterized using several tools. In powder X-ray diffraction (PXRD) pattern of Fe-L$_1$@EGO-900 (Fig. 2), a broad peak at $2\theta = 26.5°$ confirmed the presence of few layers of reduced graphene oxide support. Other inorganic phases were confirmed by the presence of diffraction peaks indexed to Fe$_3$O$_4$, Fe$_3$N, β″-Fe$_2$O$_3$, and Fe$_7$C$_3$. Formation of Fe$_7$C$_3$ and Fe$_3$N can be understood in terms of the decomposition of N- and C-rich L$_1$ in the proximity of metal atoms that aggregate to form carbide and nitride nanoparticles. Surprisingly, we could not detect the presence of Fe metal, which is possible under the synthesis conditions in the presence of a reducing agent like graphite. It is likely that smaller nanoparticles of Fe-rich phases could have oxidized to iron oxides when exposed to the ambient conditions.

The morphology of the Fe-L$_1$@EGO-900 catalyst was analyzed using scanning electron microscopy, which clearly showed graphene layers supporting spatially well-separated nanoparticles that are probably rich in iron (Supplementary Fig. 106). The microstructure of Fe-L$_1$@EGO-900 catalyst was analyzed using

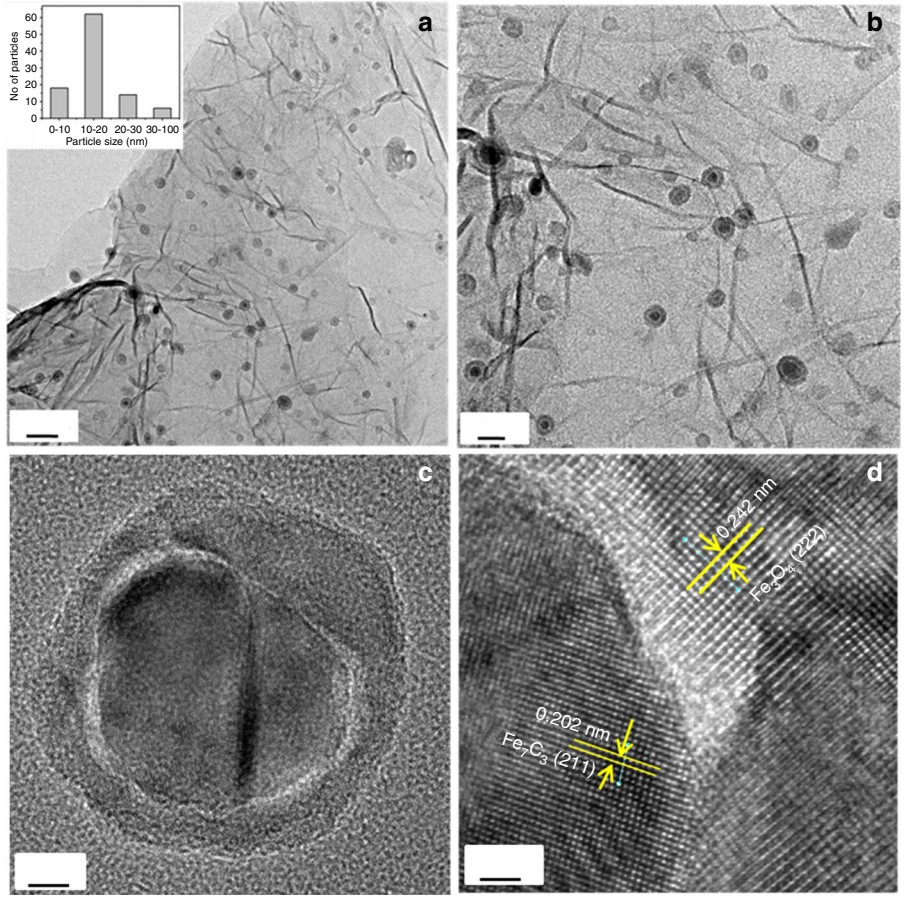

**Fig. 3** Bright-field TEM images of Fe-L$_1$@EGO-900. **a** TEM image at the scale bar 50 nm with inset showing an histogram of size of 100 nanoparticles. **b** TEM image at the scale bar 20 nm. **c** TEM image of a single nanoparticle at the scale bar 5 nm. **d** High-resolution lattice fringes of Fe$_7$C$_3$ (211) planes and Fe$_3$O$_4$ (222) planes at the scale bar 1.2 nm

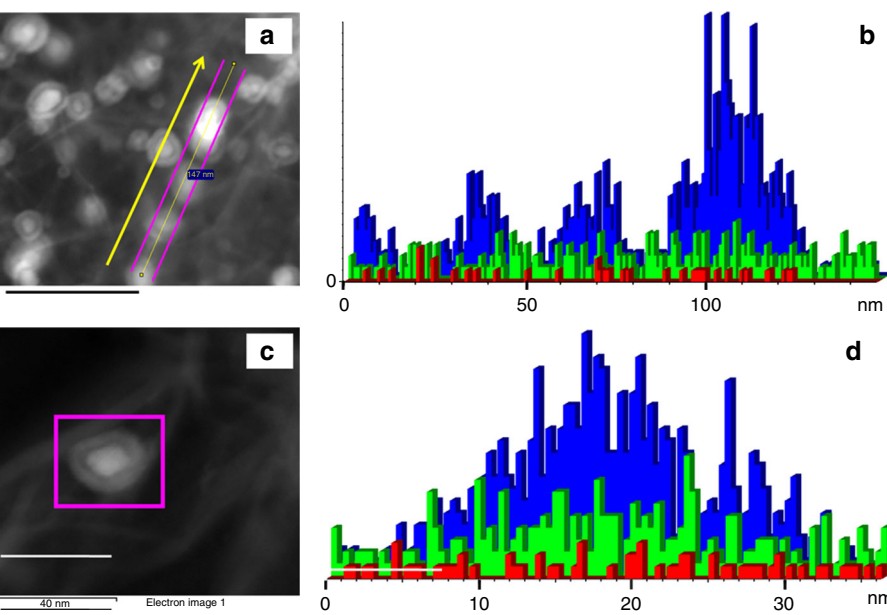

**Fig. 4** Scanning transmission electron microscope (STEM) images of Fe-L$_1$@EGO-900. **a** STEM image of Fe-L$_1$@EGO-900 catalyst. **b** Line profile of iron (blue), oxygen (green), and nitrogen (red) passing through the line. **c** STEM image of single particle. **d** Line profile of single iron nanoparticle iron (blue), oxygen (green), and nitrogen (red) passing through the particle. Scale bar is 40 nm

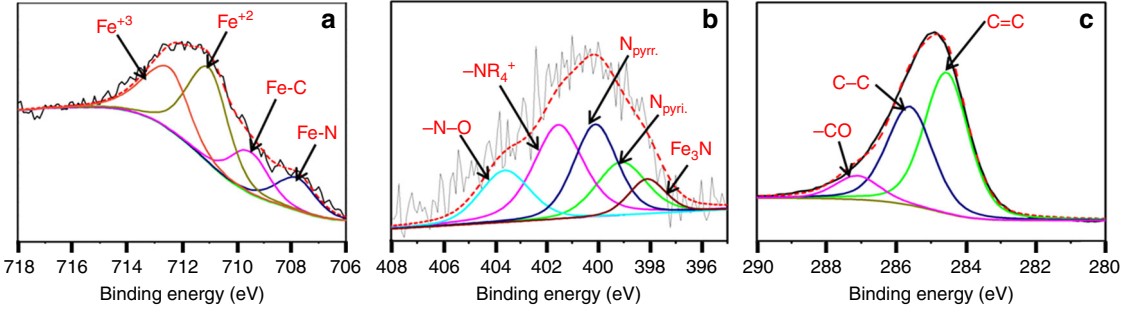

**Fig. 5** X-ray photoelectron spectra (XPS) of Fe-L$_1$@EGO-900. **a** Iron. **b** Carbon. **c** Nitrogen

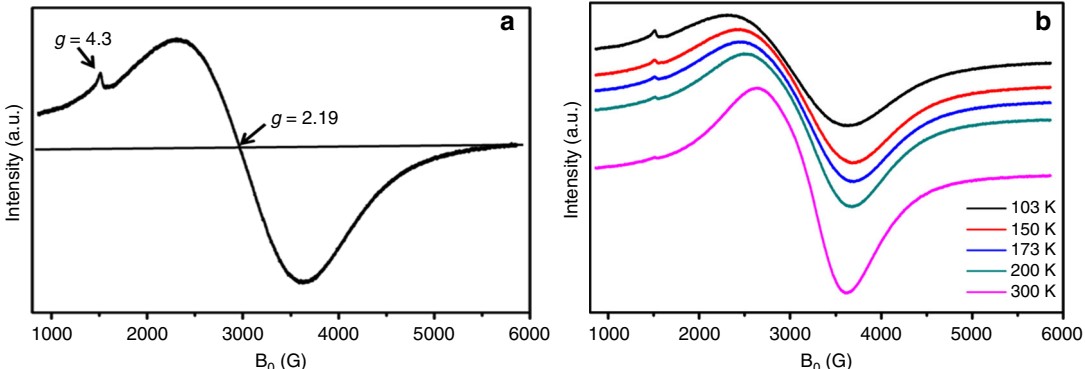

**Fig. 6** EPR of Fe-L$_1$@EGO-900 catalyst. **a** at 103 K. **b** at different temperatures

transmission electron microscopy (TEM) and the data are shown in Fig. 3. The bright-field image of Fe-L$_1$@EGO-900 showed Fe-rich nanoparticles darker in contrast compared to the graphene support. The nanoparticles were distributed throughout the graphene sheets and were of size ranging from 4 to 100 nm with majority of the particles the range of 11–20 nm as shown in the histogram (Fig. 3a). Careful analysis of bright-field TEM image suggested the particles had a core–shell architecture (Fig. 3c). High-resolution transmission electron microscopy (HRTEM) image of the sample (Fig. 3d) showed lattice fringes of 2.02 Å corresponding to the d spacing of (211) plane of Fe$_7$C$_3$ at the core and 2.42 Å corresponding to the d spacing of (222) plane of Fe$_3$O$_4$ phase. Further HRTEM image analysis of several particles revealed that the core was composed of iron carbide while the shell was dominated by iron oxides (Fig. 3d and Supplementary Fig. 107). Elemental analysis using energy dispersive X-ray analysis under STEM mode further supported the distribution of Fe across the entire particle (Fig. 4 and Supplementary Fig. 108). Importantly, the graphitic carbon shell which is commonly observed in other reported catalysts[21–23] prepared by thermal pyrolysis of molecular complex was absent in Fe-L$_1$@EGO-900.

To obtain further insight into the structure of the catalyst and especially the role of nitrogen from the organic ligand, X-ray Photoelectro Spectorscopy (XPS) investigations on the bonding of nitrogen and iron were carried out (Fig. 5). XPS studies clearly showed that nitrogen from L$_1$ was doped into the extended carbon lattice of EGO support and existed as pyridine, pyrrole, and quaternary ammonium groups. Interestingly, five distinct peaks were observed in the N1s spectra of the Fe-L$_1$@EGO-900 catalyst with the binding energy of 397.5, 399.1, 400.1, 401.4, and 403.6 eV[40,41]. The electron-binding energy of 399.1 and 400.1 eV are characteristic for pyridine-type nitrogen, and pyrrole-type nitrogen contributing two electrons to the carbon matrix. Formation of Fe$_3$N and Fe$_7$C$_3$ nanoparticles is further supported

by XPS measurement with peaks at 707.8 and 709.5 eV due to Fe–N and FeC bonds, respectively[42]. Peak at 711 eV is due to Fe$^{+2}$ and 712.45 for Fe$^{+3}$ confirms formation of Fe$_2$O$_3$ and Fe$_3$O$_4$. In Raman spectra (Supplementary Fig. 109), the observed $I_D/I_G$ increased to 1.36 in Fe-L$_1$@EGO-900 as compared to 1.13 in reduced graphitic oxide (RGO) due to increase in the disorderliness of graphene sheets caused by the deposition of iron nanoparticles in Fe-L$_1$@EGO-900. Similarly, G band of Fe-L$_1$@EGO-900 (located at 1595 cm$^{-1}$) showed a blue shift of 9 cm$^{-1}$ as compared to RGO, which may be due to charge transfer from graphene to iron nanoparticles. Energy dispersive X-ray analysis revealed an atomic ratio of C, N, O, and Fe to be 43, 1, 2.3, and 3.2, respectively (Supplementary Fig. 110 and Supplementary Table 1).

The electron paramagnetic resonance (EPR) spectra of Fe-L$_1$@EGO-900 catalyst recorded in the temperature range 103–300 K and showed an intense broad signal at g ~ 2.19 and a weak signal at g ~ 4.3 at 103 K (Fig. 6). The line width of the signal at g ~ 2.19 increased (from 985 to 1320 G) as the temperature was lowered from 300 K to 103 K. Also a marginal change in the g value (from 2.04 to 2.19) was noted at lower temperatures. These observations are indicative of the ferromagnetic behavior of Fe-L$_1$@EGO-900 containing Fe(III) nanoparticles[40]. Notably, the presence of metallic iron can be ruled out because their magnetic properties are not in line with the observed EPR intensity behavior.

In order to understand the role of ligands in the nanocatalyst formation, composition and microstructure two different iron catalysts Fe-L$_2$@EGO-900 and Fe-L$_3$@EGO-900 were prepared under the same conditions using 2,2′-bipyridine (L$_2$), and pyridine (L$_3$) as ligands, respectively. The PXRD of the catalytic materials gave an insight into the chemical composition (Supplementary Fig. 111). Notably, the chemical composition of the Fe-L$_3$@EGO-900, Fe-L$_2$@EGO-900, and Fe-L$_1$@EGO-900

**Table 2 Nanoscale iron-catalyzed acceptorless dehydrogenation of *N*-heterocycles**

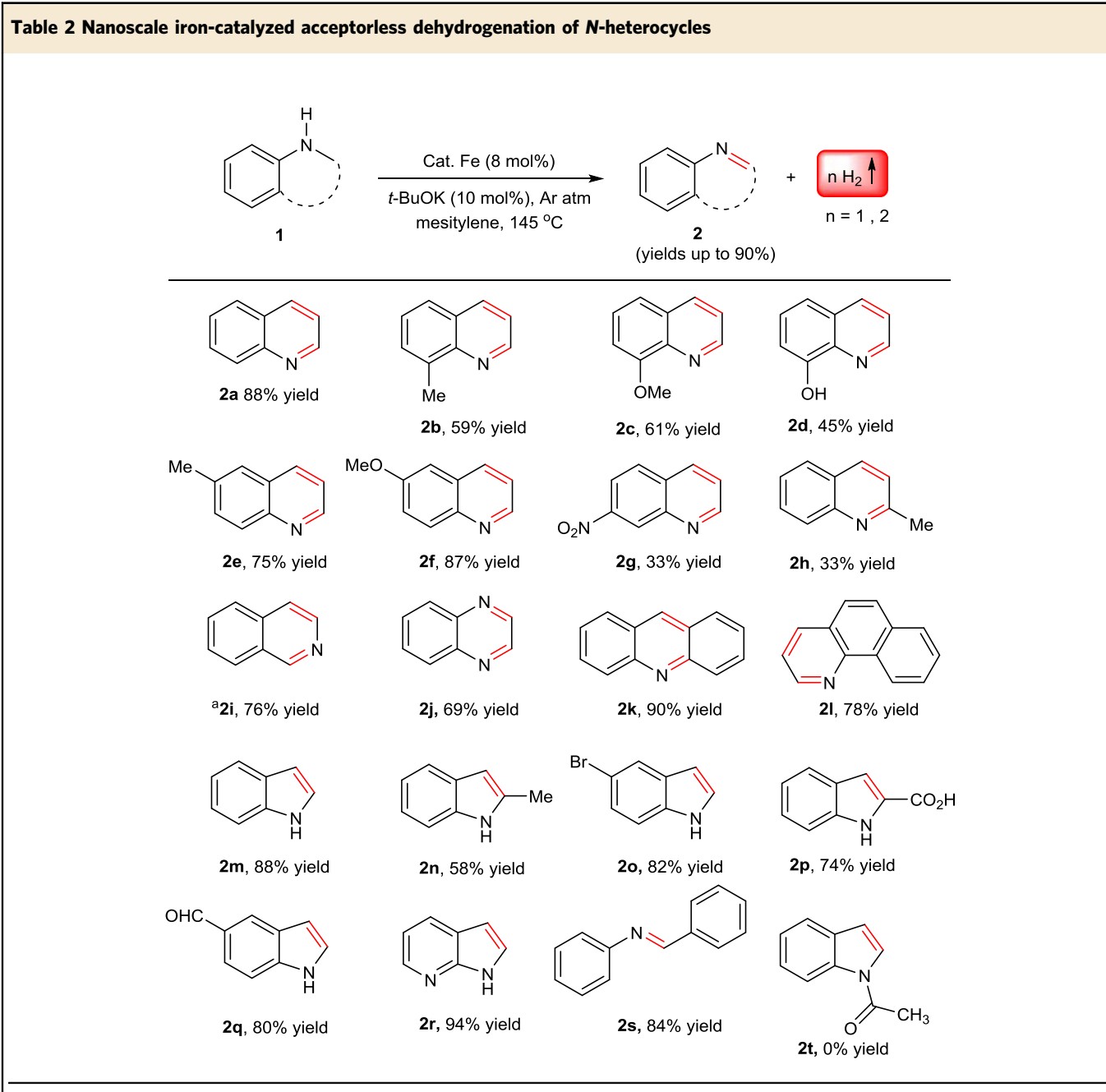

Reaction conditions: **1** (0.5 mmol), cat. Fe-L$_1$@EGO-900 (8 mol%), *t*-BuOK (10 mol%), and mesitylene (2 mL) heated at 145 °C. Yields shown are of isolated products
[a]Product 3,4-dihydroisoquinoline was observed (8%) on GC

were slightly different. In comparison to Fe-L$_2$@EGO-900 and Fe-L$_3$@EGO-900, the catalyst Fe-L$_1$@EGO-900 showed intense peaks suggesting a higher degree of crystallinity. The peaks corresponding to the oxide and the nitride phases were also higher in intensity than the carbide phase in Fe-L$_1$@EGO-900.

The structure and morphology of Fe-L$_3$@EGO-900, Fe-L$_2$@EGO-900, and Fe-L$_1$@EGO-900 were analyzed using TEM (Supplementary Fig. 112). On comparison with Fe-L$_1$@EGO-900, the microstructure of the nanoparticles synthesized using L$_2$ and L$_3$ ligands showed noticeable differences. The TEM analysis of the catalyst Fe-L$_1$@EGO-900 clearly showed the presence of spherical nanoparticles in the size range of 4–100 nm. The majority of the particles were in 15–20 nm and were supported well on the graphene support. On higher magnification, the particles showed a distinguished core–shell structure. In the case of 2,2′-bipyridine

(L$_2$), we have observed that the catalyst had iron-rich nanoparticles that were reasonably well distributed and supported on the graphene sheet in the size ranging from 30 to 300 nm (the majority size of the particles around 50 nm). On careful inspection of TEM images, nanoparticles having the core–shell morphology were seen occasionally. The catalyst precursor with pyridine (L$_3$) as a ligand produced larger nanoparticles in the range of 30–300 nm were observed. Interestingly, the nanoparticles were solid and uniform in texture without core–shell type of structure. Extensive analysis also showed not all nanoparticles were supported on the graphene sheet.

The effect of pyrolysis temperature on the composition and microstructure was studied by pyrolysing Fe-L$_1$ on EGO at various temperatures like 400, 600, and 900 °C. The PXRD patterns showed a broad peak ranging between $2\theta = 20°$ and 30°

**Fig. 7** Synthesis of precursor (**4**) for nM5- lipoxygenase inhibitor. **a** Reaction conditions: To a 20 mmol of benzyl azide (in dichloromethane) was added TfOH (1.1 equiv.) followed by 4-fluorostyrene (2.0 equiv.) at 0 °C and allowed to stir for 1 h. **b** Reaction conditions: **3** (0.2 mmol), cat. Fe-L$_1$@EGO-900 (8 mol%), t-BuOK (10 mol%), and mesitylene (2 mL) heated at 145 °C for 24 h

with characteristics of graphitic peak and an unidentified peak in all the samples (Supplementary Fig. 113). At lower pyrolysis temperatures, there was overlapping of peaks in this region which could be due to the presence of poorly crystalline carbonaceous materials. Importantly, Fe-L$_1$@EGO-900 that was prepared by pyrolyzing at 900 °C showed a single sharp peak corresponding to graphitic basal plane at $2\theta = 26.5°$ in addition to clear peaks due to crystalline phases of oxides, nitrides, and carbides. From PXRD data and TGA (Supplementary Figs. 113 and 114), it was clear that high pyrolysis temperature (~900 °C) resulted in the decomposition of the iron-ligand complex to from iron oxide, nitride, and carbide phases. The effect of pyrolysis temperature on the distribution of size and morphology of the nanoparticles was studied using TEM (Supplementary Fig. 115). The catalyst Fe-L$_1$@EGO-400 showed well distributed small iron nanoparticles in the size range of 2–10 nm. The individual nanoparticles were not in specific shape and no core–shell structure was observed. As the temperature was increased from 400 to 600 °C, a few particles agglomerated to form bigger particles with an average size of 10–12 nm. Both PXRD (Supplementary Figs. 111 and 113) and TEM images (Supplementary Figs. 112 and 115) strongly indicated that the choice of ligands and pyrolysis temperature affected the composition, size distribution, and microstructure of the individual nanoparticles in the catalyst.

**Initial investigations in acceptorless dehydrogenation reactions**. All the prepared materials were tested for their reactivity toward dehydrogenation of N-heterocycles with the concomitant generation of dihydrogen under oxidant- and acceptor-free conditions. We began our initial study using 1,2,3,4-tetrahydroquinoline (**1a**) as a benchmark substrate. The effect of each of the key parameters such as type of support, nitrogen ligands, Fe:ligand molar ratios, and pyrolysis temperature were carefully investigated (Table 1). As seen from Table 1, in situ generated homogeneous metal complex, as well as carbon-supported non-pyrolyzed materials did not show any catalytic activity under optimal conditions (Table 1, entries 1–2). Similarly, pyrolyzed iron salt (uncomplexed to L$_1$), and pyrolyzed L$_1$ ligand on carbon support were also found to be ineffective (Table 1, entries 3–4). We have observed that the iron–phenanthroline (Fe-L$_1$) complex pyrolyzed with exfoliated graphitic oxide at 900 °C (Fe-L$_1$@EGO-900) led to a highly active catalytic material for dehydrogenation of **1a** (Table 1, entries 5–7). Under the optimized reaction conditions, the catalyst Fe-L$_1$@EGO-900 selectively formed quinoline (**2a**) in 88% isolated yield with the complete conversion of **1a** (Table 1, entry 7). The generation of molecular hydrogen was qualitatively analyzed by gas chromatography (Supplementary Fig. 116). Other pyrolyzed carbon-supported iron catalysts using related nitrogen ligands, such as 2,2′-bipyridine (L$_2$) and pyridine (L$_3$), also showed activity yielding the dehydrogenated product

(**3a**) in low yields (Table 1, entries 8–9). It was worth noting that under similar experimental conditions, catalysts Fe-L$_1$ prepared on other conventional supports such as Al$_2$O$_3$, SiO$_2$, CeO$_2$ and TiO$_2$ showed no activity in the AD with the release of H$_2$ (Table 1, entries 11–14). The iron catalyst was easily separated from the reaction medium under the external magnetic field (Supplementary Fig. 117).

**Acceptorless dehydrogenation of N-heterocycles**. The catalytic dehydrogenation of N-heterocycles with concomitant generation of dihydrogen constitutes is not only a fundamental process in synthetic chemistry, but also have direct applications in energy storage systems such as liquid organic hydrogen storage materials and fuel cells[43–45]. Despite several homogeneous catalysts based on iridium[46–49], Ru–hydride complexes[2,12], Co–pincer complex[50] were successfully employed in the catalytic dehydrogenation of N-heterocycles without using hydrogen acceptor and additives, there is only one example of an iron-based catalytic system known so far (Fig. 1b)[51]. However, the development of stable, inexpensive, reusable heterogeneous system[52] for the dehydrogenation of N-heterocycles operates under practical conditions with the liberation of hydrogen gas is highly desirable and very demanding.

Inspired by these literature reports and with the developed efficient system for AD strategy in hand (Table 1), we sought to apply for oxidant-free, acceptorless dehydrogenation of various N-heterocyclic compounds. The expected, Fe-L$_1$@EGO-900 catalyst material displayed the best activity in the complete dehydrogenation of N-heterocycles with the liberation of molecular hydrogen. As shown in Table 2, a number of partially saturated N-heterocycles containing secondary amines were successfully dehydrogenated into corresponding N-heteroaromatic compounds with extrusion of dihydrogen. To our delight, almost all N-heterocyclic scaffolds for, e.g., 1,2,3,4-tetrahydroquinoline (**1a**–**1h**), 1,2,3,4-tetrahydroisoquinoline (**1i**), 1,2,3,4-tetrahydroquinoxaline (**1j**), 1,2,3,4-tetrahydrobenzo[h]quinoline (**1l**), indoline (**1m**–**1q**), and 2,3-dihydro-1H-pyrrolo[2,3-b]pyridine (**1r**) were completely dehydrogenated under acceptorless conditions. Importantly, bromo, formyl, carboxyl, hydroxyl, methoxy, and nitro-groups were well tolerated under the optimized conditions and yielded the desired products in moderate to good yields. In order to demonstrate the stability and reusability, the active catalyst was successfully recycled for four times without any significant loss of activity for a model substrate (Supplementary Fig. 118). These results evidently demonstrate that our catalyst is highly stable and robust. In addition, the catalyst is conveniently handled under ambient atmosphere.

The high reactivity of Fe-L$_1$@EGO-900 catalyst toward the acceptorless dehydrogenation of tetrahydroquinoline derivatives encouraged us to extend its application in the synthesis of

**Table 3 Iron-catalyzed dehydrogenation of amines to imines with extraction of H$_2$**

Reaction conditions: **5** (0.5 mmol), cat. Fe-L$_1$@EGO-900 (8 mol%), t-BuOK (10 mol%), and mesitylene (2 mL) heated at 145 °C
[a]Yields of **6** was determined by GC using m-xylene as an internal standard. Yields shown within brackets are of isolated products

pharmaceutically important molecules. Thus, dehydrogenation of **3** selectively yielded **4**, a precursor for nM5-lipoxygenase inhibitor (Fig. 7).

Given the observed reactivity of the present iron-catalyzed dehydrogenation of N-heterocycles, we were interested in getting insights into the reaction mechanism (Supplementary Fig. 119). Notably, no dehydrogenation product was detected when using 1-(indolin-1-yl)ethan-1-one (**1t**) as substrate. This result clearly indicated that the presence of the N–H motif in the heterocyclic compound is very critical for the acceptorless dehydrogenation reaction, which might be favorable for the adsorption of partially saturated N-heterocycles onto the catalyst surface. It was observed that a superoxide radical anion quencher (butylated hydroxytoluene, BHT) had no effect on the reaction rate in the formation of **2a**, thus the involvement of radical mechanism was completely ruled out. It is well known that BHT suppresses the formation of a superoxide radical anion (due to the presence of residual oxygen/air or to the use of t-BuOK). Indeed, a catalytic amount of base is required to activate the substrates.

**Quantification of dihydrogen**. Acceptorless dehydrogenation of 1,2,3,4-tetrahydroquinoline (**1a**) to quinoline (**2a**) is accompanied by the concomitant release of two molecules of hydrogen. The generation of molecular hydrogen was qualitatively analyzed by gas chromatography (Supplementary Fig. 116). The generated hydrogen was quantified for the model experiment and the reported wt% of hydrogen is calculated based on the substrate, not taking into consideration the solvent. A volumetric quantitative analysis of 1,2,3,4-tetrahydroquinoline dehydrogenation revealed that ~81% formation of dihydrogen (Table 1, entry 7 and Supplementary Fig. 120). In another experiment, the AD reaction was conducted in a flask that was connected through a rubber tube to a

second flask in which cyclohexene and a catalytic amount of Wilkinson's catalyst in benzene were placed. After the reaction was completed, cyclohexane was produced in 45% yield in the second flask (after 20 h, yield of **2a** is 61%), demonstrating that the hydrogen gas generated in the AD reaction is responsible for hydrogenation of cyclohexene (Supplementary Figs. 121 and 122). This dual reaction involving hydrogenation of alkene authenticated that hydrogen is produced during the course of the reaction.

**Acceptorless dehydrogenation of amines to imines**. Imines serve as a key intermediate for the preparation of fine chemicals, pharmaceuticals, and natural products[53]. Traditional methods to access imines involve a condensation reaction of the amine with active carbonyl compounds and often require dehydrating agents as well as Lewis acid catalysts. In a complementary approach, the catalytic dehydrogenative coupling reaction of an alcohol with an amine offers the direct formation of imine derivatives. Nevertheless, this reaction operates at elevated temperature with challenges associated to the competing hydrogen autotransfer (HA) strategy (in situ hydrogenation of imine to afford the amine)[54]. Indeed, acceptorless dehydrogenation of amine to imine by precious metals provides an oxidant-free strategy with extrusion of dihydrogen as a sole by-product. However, efficient catalytic methods for the direct conversion of primary amines to imines are challenging, due to presence of an α-hydrogen in the intermediate imine, which may be rapidly dehydrogenated to a nitrile and the formation of a mixture of secondary amine (N-alkylated amine) and imine products[54]. Recently, the selective synthesis of imines from nitroarenes and aldehydes or ketones through hydrogenation has been reported using a reusable cobalt catalyst[55]. These insights are put to use our Fe-based catalytic system for the direct conversion of an amine into imine under

**Table 4 Nanoscale iron-catalyzed acceptorless dehydrogenation of primary alcohols**

Reaction conditions: Primary alcohol **8** (0.5 mmol), cat. Fe-L$_1$@EGO-900 (8 mol%), *t*-BuOK (10 mol%), and n-octane (2 mL) heated at reflux under open argon atm. Yields shown are of isolated products
[a]Yields are based on GC

more economical and environmentally benign conditions (Table 3).

Various substituted benzylamines (**5a–5e**) and pyridin-2-ylmethanamine (**5g**) were dehydrogenated under optimized conditions and selectively gave the desired imines in good yield. Notably, more sterically hindered benzylamines such as *ortho*-substituted and α-substituted benzylamine were also dehydrogenated (**6f** and **6h**). To our delight, aliphatic amine (**5i**) gave both imine and doubly dehydrogenated product (nitrile compound **7**) in 1:1 ratio under our conditions.

**Acceptorless dehydrogenation of primary alcohols.** Casey and co-workers reported the first iron-catalyzed dehydrogenation of alcohols using the Knölker complex, (hydroxycyclopentadienyl)

iron dicarbonyl hydride **I**, as a catalyst in the presence of hydrogen acceptor (Fig. 1a)[56]. Regrettably, the scope of the reaction was limited only to secondary alcohols and conjugated primary alcohols. In 2014, dehydrogenation of 2-pyridylmethanol derivatives catalyzed by an iron–cyclopentadienyl complex **II** (Fig. 1a) has been reported under acceptorless conditions[57]. Of late, Hong and co-workers developed in situ generated iron catalytic system (a combination of Fe(III) salt, 1,10-phenanthroline, and K$_2$CO$_3$) for dehydrogenation of secondary benzyl alcohols to afford the corresponding ketones[58]. However, other unactivated secondary alcohols and primary alcohols were ineffective, and no dehydrogenation was observed. Based on metal–ligand cooperation mechanism, a PNP-iron pincer complex (**III**) catalyzed acceptorless dehydrogenation of alcohols to esters has been

**Table 5 Acceptorless dehydrogenation of secondary alcohols and a diol**

11a, 97% yield

11b, 98% yield

11c, 95% yield

11d, 90% yield

11e, 73% yield

11f, 77% yield

11g, 71% yield

11h, 80% yield

11i, 69% yield

11j, 72% yield

11k, 43% yield

11l, 54% yield

11m, 36% yield

11n, 51% yield

11o, 40% yield

13, 97% yield[a]

Reaction conditions: Alcohol **10** (0.5 mmol), cat. Fe-L$_1$@EGO-900 (8 mol%), $t$-BuOK (10 mol%), and n-octane (2 mL) heated at reflux under open argon atm. Yields shown are of isolated products
[a]1,2-phenylenedimethanol (**12**) was used

reported by the research group of Jones and Schneider (Fig. 1a)[59]. Notably, primary alcohols underwent self-dehydrogenative coupling to form esters presumably via Tishchenko reaction. To the best of our knowledge, there have been no reports on reusable, robust iron-based catalytic system for AD strategy of a series of various alcohols under mild conditions.

Herein, we disclose an efficient, reusable iron-catalyzed oxidant-free, acceptorless dehydrogenation of alcohols such as primary alcohols to aldehydes, secondary alcohols to ketones, and diol to lactone (Tables 4 and 5). After having demonstrated the excellent activity of the Fe-L$_1$@EGO-900 catalyst in the model reaction (Table 1), we have investigated the scope of a series of structurally diverse benzylic alcohols in the acceptorless dehydrogenation. As shown in Table 4, benzylic alcohols containing electron-donating as well as electron-withdrawing groups are compatible with the catalytic system and good to excellent yields of the corresponding aldehydes were obtained selectively under very mild conditions (up to 94%). The generation of molecular hydrogen was qualitatively analyzed by gas chromatography (GC). Notably, no generation of CO was observed and this is an important point regarding use in fuel cell applications. Interestingly, bis-aldehyde (**9i**) can be obtained in a straightforward manner in up to 70% yield. A more sensitive cinnamyl alcohol underwent dehydrogenation and efficiently gave the corresponding aldehyde **9j** in moderate yield (69%) under open argon atmosphere. To our delight, substrates with phenoxy, and amine groups on the aromatic ring also afforded the desired products in good yields (products **9m** in 69% and **9n** in 61% yields, respectively). The biomass-derived furfuryl alcohol was readily converted into the corresponding aldehyde in good yield (product **9o** in 55% yield). Under optimal conditions, 1-hexanol showed poor reactivity and yielded 1-hexanal (**9p**) in 21% yield. It is important to note that the selective conversion of alcohols to aldehydes via AD startegy is very challenging due to competing ester formation[2,60,61].

**Scope of secondary alcohols and diol**. We examined the scope of secondary alcohols in the catalytic AD strategy using Fe-$L_1$@EGO-900 catalyst (Table 5). Secondary benzylic alcohols are dehydrogenated to the corresponding acetophenone derivatives in good isolated yields (up to 98%). The reaction is tolerant to a variety of functional groups such as –OMe, –Me, and –$NO_2$ as well as halides (–Cl and –Br). Electronic influence on the dehydrogenation activity seemed to be significant because a substrate containing an electron-donating –OMe group (product **11h** in 80%) gave excellent yield than the one with an electron-withdrawing –$NO_2$ group (product **11i** in 69%). Importantly, the homologous of 1-phenylethan-1-ol (i.e., 1-phenylpropan-1-ol) gave moderate yield under optimal conditions (products **11k** in 43% and **11e** in 73% yield). In addition to aromatic substrates, the aliphatic secondary alcohol cyclohexanol was successfully dehydrogenated to give cyclohexanone (**11o**) in 40% yield. A substrate with 5- and a 6-membered ring containing 2° alcohols almost gave same yields (products **11l** and **11n** in 54% and 51% yields, respectively). Besides primary alcohols, and secondary alcohols, diol was also examined. Accordingly, dehydrogenation of 1,2-benzenedimethanol (**12**) gave 100% conversion and complete selectivity for the corresponding lactone **13** by releasing two equiv. of dihydrogen. However, the corresponding 1,4-disubstituted derivative gave the corresponding 1,4-dialdehyde derivatives **9i** under the same reaction conditions. Perhaps, intramolecular dehydrogenative condensation is more favorable in case of 1,2-disubstituted primary diols and readily produces the corresponding lactone[61, 62].

**Microstructure and catalytic activity of the nanocatalysts**. The composition of the catalyst and its structural arrangement within a nanospace is strongly correlated with the exceptional activity for acceptorless dehydrogenation reaction. The composition of the catalyst and pyrolysis temperature have been shown to play an important role in the formation and the activity of Fe-$L_1$@EGO-900, the most active catalytic material for dehydrogenation reactions. It can be concluded that iron precursor using rigid ligands such as $L_1$ formed nanoparticles with a narrow size distribution. In addition, the increased π area of the ligand also ensured favorable molecular orientation of the precursor complex with the EGO support. Consequently, after pyrolysis the uniformity in distribution of the nanoparticles was high. These factors were less efficient in the case catalysts prepared from $L_2$ to $L_3$ ligands as can be seen from their respective TEM images and the catalytic activity (Supplementry Fig. 112 and Table 1). The mechanism of formation of core–shell structure in the case of Fe-$L_1$@EGO-900 and to some extent in Fe-$L_2$@EGO-900 is a subject of in-depth study. However, one can suspect that segregation of individual phases of iron into separate particles was less favored with increasing rigidity of the ligand. Thus, Fe-$L_1$@EGO-900 and Fe-$L_2$@EGO-900 formed a core–shell structure while Fe-$L_3$@EGO-900 formed separate spherical nanoaparticles although the chemical compositions as observed from PXRD (Supplementry Fig. 111) were similar in all the three catalytic materials. This was a unique departure from the catalysts reported earlier[21–25] where thermal pyrolysis of metal–ligand complexes invariably yielded spherical nanoparticles encapsulated in a sheath of graphitic carbon. Besides, the carbon shell may result in a poor accessibility (for the reactants) to the sites for a new reaction pathway.

The catalytic activity differed with the microstructure of the nanocatalyst in a significant manner. The reason can be ascribed to the type of carbon support used in the present work. The interaction of the carbon support with the catalyst precursor is likely to have played a pivotal role in the kinetics of decomposition, nucleation, surface migration, and agglomeration

finally forming a nanocatalyst. Careful observation shows that the core of Fe-$L_1$@EGO-900 is composed of carbides while the shell is composed of mixture of oxides. The observation points out to the scenario, wherein iron carbides formed in the initial stages of nucleation followed by growth mediated by the aggregation of iron-rich clusters that migrated on the EGO. The composition of the iron-rich clusters is dependent on the nature of the ligand, Fe:ligand ratio in the complex precursor. Equally important is the rate of aggregation of the iron clusters forming the nanocatalysts, which may be highly influenced by the surface of the EGO support. Certainly, the surface chemistry of the EGO and Vulcan carbon (commonly used in earlier reports)[21–25] are different and therefore play different role in the nucleation and growth of the nanocatalysts. The difference has led to different microstructure and catalytic activity. However, in-depth studies involving temperature-based surface studies involving STM could throw more light into the mechanism of formation of different microstructures. The activity of the nanocatalysts in the dehydrogenation reaction clearly showed Fe-$L_1$@EGO-900 gave better yield. Our efforts to produce a catalyst using Vulcan carbon support did not yield an active catalyst. The control experiments showed neither EGO, nor N-doped EGO nor pure phases of iron oxides, carbides, and nitrides (Supplementary Table 2) showed a high level of conversion and selectivity as that of Fe-$L_1$@EGO-900 pointing out to the role of interfaces of mixed phases of iron.

## Discussion

A unified strategy for oxidant-free and acceptorless dehydrogenation reactions of relatively abundant alcohols such as primary alcohols to aldehydes, secondary alcohols to ketones, diol to lactone, and N-heterocycles with the concomitant generation of hydrogen gas catalyzed by iron-based nanocatalyst is described. The catalytic material described in this work was obtained by thermally pyrolyzing Fe(acac)$_3$:phenanthroline complex on exfoliated graphitic oxide support with a unique core–shell architecture composed of oxide as a shell and carbide as a core of iron without the encapsulating sheath of carbon. This elegant system offers a new streamlined strategy for the sustainable production of chemicals with a great step-economy and reduced waste generation. In addition to the interesting correlation of microstructure and the catalytic activity of this inexpensive and reusable catalyst, the work also highlights the greenness of the reaction due to the formation of hydrogen gas as the only by-product.

## Methods

**Synthesis of Fe-$L_1$@EGO-900 catalyst**. In a 100 mL beaker Fe(III) acetylacetonate precursor (0.5 mmol) and 1,10-phenanthroline ligand (0.5 mmol) were dissolved in 30 mL of ethanol and sonicated for 2 h to form Fe–phenanthroline complex. In another 250 mL beaker, 560 mg of EGO support was taken in 70 mL of ethanol and sonicated for 2 h. The above-obtained EGO suspension and Fe–phenanthroline complex solution were mixed together in 250 mL beaker and further sonicated for 2 h. The suspension was refluxed at 85 °C for 4 h and after cooling down to room temperature ethanol was evaporated in vacuum. The solid sample obtained was dried at 80 °C for 14 h. Then, it was ground to a fine powder followed by calcination at 900 °C under a stream of argon with the flow rate of 30 mL min$^{-1}$ and the heating rate: 25 °C min$^{-1}$ for about 4 h to obtain a catalyst Fe-$L_1$@EGO-900. ICP-AES analysis was done to determine the amount of iron present and was found to be 5.32%.

**General procedure for acceptorless dehydrogenation of N-heterocycles**. To an oven-dried schlenk tube (25 mL), Fe-$L_1$@EGO-900 catalyst (47 mg, 8 mol%), t-BuOK (10 mol%), N-heterocycles (0.5 mmol), and mesitylene (2 mL) were added under argon atmosphere. The solution was heated at 145 °C with stirring under open argon flow for 18–24 h. After cooling down the reaction mixture to room temperature, the catalyst was separated from the reaction mixture by centrifugation and the reaction mixture was analyzed by GC and GC-MS. The supernatant was transferred into another flask, and the catalyst was washed with EtOAc (2 × 4 mL) and the washings were collected. The solvent was evaporated from the reaction

mixture, and the crude product was subjected to silica gel column chromatography using EtOAc: petroleum ether to afford the product.

**General procedure for acceptorless dehydrogenation of alcohols**. To an oven-dried schlenk tube (25 mL), Fe-L$_1$@EGO-900 catalyst (47 mg, 8 mol%), t-BuOK (10 mol%), alcohol (0.5 mmol), and n-octane (2 mL) were added under argon atmosphere. The solution was refluxed with stirring under open argon flow for 16–24 h. After cooling down the reaction mixture to room temperature, the catalyst was separated from the reaction mixture by centrifugation and the reaction mixture was analyzed by GC and GC-MS. The supernatant was transferred into another flask, and the catalyst was washed with EtOAc (2 × 4 mL) and the washings were collected. The solvent was evaporated from the reaction mixture, and the crude product was subjected to silica gel column chromatography using EtOAc: petroleum ether to afford the corresponding carbony compound.

**Data availability**. The authors declare that the data supporting the findings of this study are available within the article and its Supplementary Information files. All other relevant data are available from the authors on reasonable request.

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

## Acknowledgements

This research was supported by the SERB (SB/FT/CS-065/2013) and CSIR-NCL. G.J. thank UGC and V.G.L. thank CSIR for fellowship. D.J. acknowledges the financial support of Ramanujan Fellowship (RJN-112/2012). E.B. thank Dr C. S. Gopinath and Dr C. P. Vinod for XPS analysis, Dr D. Srinivas for EPR interpretation, Dr. B. L. V. Prasad and Dr. C. V. Ramana for their helpful discussion. We also thank IIT-Bombay (EPR analysis), IACS-Kolkata (STEM), and JNCASR-Bangalore (HRTEM) for analytical support.

## Author contributions

G.J.: Catalysts synthesis, characterization, catalytic experiments, mechanistic studies, and contributed to manuscript writing. V.G.L.: Catalytic experiments and starting material synthesis. D.J.: Material characterization, and contributed to manuscript writing. E.B.: Design and direction of the project, conceived and supervised the project, and wrote the manuscript.

## Additional information

**Competing interests:** The authors declare no competing financial interests.

