## [Peer Review File · Nature Communications]

Reviewer #1 (Remarks to the Author):

This is a valuable addition to the literature and one that has strong practical application possibilities rather than opening up any new reaction pathways because the latter are known with other catalysts. The probable catalyst seems to be an unexpected core-shell nanoparticle with 2 advantages: recyclability and cheap starting materials. This is worth publication.

Reviewer #2 (Remarks to the Author):

The authors report on a reusable iron catalyst and application of it in the acceptor-less dehydrogenation of N heterocycles, amines and alcohols. I feel that the work has the potential for a top chemistry journal based on the broad applicability of the reusable catalyst.

Strengths: Reusable Fe catalyst with an indeed very broad scope. After the impressive work in which colleagues have reported on homogeneous base metal catalysts for diverse reactions, the trend goes certainly towards reusable and easy to handle Co, Fe and Mn catalyst. The manuscript is not initiating this trend but extends it towards important dehydrogenation reactions.

Weaknesses: the catalysts design work is close to that reported by the Beller group and the amine dehydrogenation leads to uninteresting products. The catalyst consists of rather large Fe-containing particles and a lot of particle phases have been observed.

It is difficult to see/accept what the actual catalyst is and I feel more work is needed to address this question.

I feel a proper citation of reusable Co and Fe (de)hydrogenation catalyst is needed. The hydrogen economy part of the introduction guides the readers a bit into a wrong direction.

Reviewer #3 (Remarks to the Author):

The results, demonstrating acceptorless dehydrogenation with mixed iron oxide/carbide-based nanoparticles, obtained by pyrolyzing an Fe(III) phenanthroline complex on an exfoliated graphitic support, are important and of broad interest. The catalyst has been well characterized and dehydrogenation demonstrated with a variety of amine and alcohol substrates. The experimental facts are adequately presented but what is missing is a good discussion of these results. The "Discussion" part consists of four sentences which are really a summary of the paper. #What is needed is a clear and adequate discussion of the experimental results. Why is phenanthroline better than other ligands in the catalyst precursor and what exactly is the difference between the catalyst obtained by treatment at 900 degrees C and those at other temperatures. In the "discussion" the authors refer to an "interesting correlation of the microstructure and catalytic activity". Undoubtedly remarks can be found buried somewhere in the "Results" section but there needs to be a discussion in the "discussion" section.

The manuscript also contains several typos which need to be removed, e.g. expoliation vs exfoliation.

Response to reviewer's comments

Referee #1:

This is a valuable addition to the literature and one that has strong practical application possibilities rather than opening up any new reaction pathways because the latter are known with other catalysts. The probable catalyst seems to be an unexpected core-shell nanoparticle with 2 advantages: recyclability and cheap starting materials. This is worth publication.

Response: We sincerely appreciate Referee #1 for the very positive recommendation on the manuscript.

Referee #2:

The authors report on a reusable iron catalyst and application of it in the acceptor-less dehydrogenation of *N*-heterocycles, amines, and alcohols. I feel that the work has the potential for a top chemistry journal based on the broad applicability of the reusable catalyst.

Strengths: Reusable Fe catalyst with an indeed very broad scope. After the impressive work in which colleagues have reported on homogeneous base metal catalysts for diverse reactions, the trend goes certainly towards reusable and easy to handle Co, Fe, and Mn catalyst. The manuscript is not initiating this trend but extends it towards important dehydrogenation reactions.

Response: We greatly appreciate the reviewer #2 for highlighting the importance and interesting chemistry of our manuscript.

Weaknesses: the catalysts design work is close to that reported by the Beller group

Response: The procedure to decompose a molecular complex of a transition-metal on an inert support has been commonly used to prepare supported nanoparticles for various applications including catalysis. We have cited the appropriate references. However, we would like to clarify that the microstructure of our nanocatalyst is strikingly different from that reported by the Beller group. The nanoparticles reported by the Beller group and others were frequently surrounded by a graphitic carbon shell. However, in our case carbide@oxide core-shell nanoparticles were

observed without any coating of graphitic carbon. The core is composed of intermixed phases of iron oxide, iron nitride, and iron carbide while the shell is composed of iron oxides that were clearly identified using high-resolution TEM images. The major difference arises from the mechanism of formation of nanoparticles that are highly dependent on the physical and chemical properties of the support.

and the amine dehydrogenation leads to uninteresting products.

Response: Kindly allow us to disagree with this comment.

Imines are key intermediates in the synthesis of fine chemicals and numerous biologically active compounds. Efficient catalytic methods for the direct conversion of primary amines to imines under mild conditions are challenging,

- 1) the generated imines are rapidly dehydrogenated to nitriles (due to presence of second α -amino hydrogen in the intermediate imine),
- 2) competing hydrogen atom transfer reaction (leads to a mixture of amine and imine products).

The catalyst consists of rather large Fe-containing particles and a lot of particle phases have been observed. It is difficult to see/accept what the actual catalyst is and I feel more work is needed to address this question.

Response: We agree with referee #2 comments. PXRD of the catalytic material shows oxides, nitride, and carbide of iron. We have performed several control experiments under optimal conditions with independently prepared pure phases of iron to confirm their role in the catalytic dehydrogenation reaction (Table S2).

Table S2 illustrates the conversion of 2-chlorobenzyl alcohol (**8c**) and the selectivity of 2-chlorobenzaldehyde (**9c**) using different Fe-based catalysts in pure phases.

Entry	Catalyst	Conversion (%)	Selectivity (%)
1	Fe-L1@EGO-900	95	93
2	Fe ₃ O ₄	15	55
3	Fe ₂ O ₃	38	40
4	Fe _x N	15	80
5	Fe ₃ C	12	70

From Table S2, it is clear that the Fe-L1@EGO-900 with mixed phases in core-shell morphology shows excellent conversion and selectivity. Other pure distinct phases of iron are not as active for this catalysis. Hence, the mixed phases of iron having specific core-shell morphology are necessary for the superior activity.

I feel a proper citation of reusable Co and Fe (de)hydrogenation catalyst is needed.

Response: Thanks. Relevant references are cited in the revised manuscript.

The hydrogen economy part of the introduction guides the readers a bit into a wrong direction.

Response: We have tried our very best to provide an informative manuscript to the general audience. We trust in the Editorial Board/Publisher to further improve it, if it should be required.

Referee #3:

The results, demonstrating acceptorless dehydrogenation with mixed iron oxide/carbide-based nanoparticles, obtained by pyrolyzing an Fe(III) phenanthroline complex on an exfoliated graphitic support, are important and of broad interest. The catalyst has been well characterized and dehydrogenation demonstrated with a variety of amine and alcohol substrates. The experimental facts are adequately presented but what is missing is a good discussion of these results. The "Discussion" part consists of four sentences which are really a summary of the paper. What is needed is a clear and adequate discussion of the experimental results.

Response: We appreciate the referee 3's comments and thank for highlighting the importance and interesting chemistry of our manuscript.

The manuscript was prepared according to the journal format ("Discussion" section is usually the summary of the paper). However, in the revised manuscript we have discussed various aspects of the catalytic material and an adequate data has been included in the ESI.

Why is phenanthroline better than other ligands in the catalyst precursor?

Response: Thanks. As per referee's suggestion, we have performed several control experiments including the effect of various ligands.

Figure S5. (i) XRD of Fe-L3@EGO-900, (ii) Fe-L2@EGO-900, and (iii) Fe-L1@EGO-900 catalyst.

To understand the structural composition of the nanocatalyst obtained by chelating iron metal with different ligands, PXRD data were taken for Fe-L3@EGO-900, Fe-L2@EGO-900 and Fe-L1@EGO-900 catalyst prepared according to the procedure described in the manuscript and the data is shown in Figure S5. The chemical composition of the Fe-L3@EGO-900, Fe-L2@EGO-900 and Fe-L1@EGO-900 was slightly different. In comparison to Fe-L2@EGO-900 and Fe-L3@EGO-900, the catalyst Fe-L1@EGO-900 showed intense peaks suggesting a higher degree of crystallinity. The peaks corresponding to the oxide and the nitride phases were slightly higher in intensity than the carbide phase in Fe-L1@EGO-900.

The structure and morphology of Fe-L3@EGO-900, Fe-L2@EGO-900 and Fe-L1@EGO-900 were analyzed using TEM and are shown in the Figure S6.

Figure S6. a & b) TEM images of Fe-L3@EGO-900 at the scale bar of 200 nm and scale bar of 20 nm; **c & d)** TEM images Fe-L2@EGO-900 at the scale bar of 200 nm and the scale bar of 20 nm; **e, f, g & h)** TEM images of Fe-L1@EGO-900 at the scale bar of 100 nm, scale bar 50 nm, scale bar of 20 nm and scale bar of 5 nm.

Figure S6 (a & b) clearly showed that the catalyst precursor with pyridine as a ligand (L3) forms larger nanoparticles having a size in the range of 30-300 nm and the agglomeration of nanoparticles were commonly observed. On careful observation, we have seen that nanoparticles were solid and uniform in nature and indeed, no core-shell type of morphology was observed (Figure S6 b). In the case of bipyridine ligand (L2), we have observed that the catalyst had iron-rich nanoparticles that were reasonably well-distributed and supported on the graphene sheet in the size ranging from 30-300 nm. Most of the nanoparticles were around 50 nm as shown in Figure S6 c. On careful inspection of TEM images, nanoparticles having the core-shell morphology were seen occasionally (Figure S6 d). The microstructure of the nanoparticle synthesized using 1,10-phenanthroline ligand (L1) showed a remarkable difference. The spherical nanoparticles in the size range of 4-100 nm (the majority of the particles were in 15 – 20 nm in size) had a well-defined core-shell structure and good particle-support interaction, as shown in Figure S6 e, f, g, & h.

The following conclusions are drawn from our observations.

- a) Iron precursor using more rigid ligands such as L1 formed nanocatalysts with a narrow distribution in particle size.
- b) In addition, the increased π area of the ligand also ensured good distribution of the precursor iron-complex on the graphene oxide support. Consequently, after pyrolysis more uniform distribution of the nanocatalysts was obtained for L1 and L2.

c) The segregation of individual phases of iron into separate particles was less favored with increasing rigidity of the ligand. Thus, Fe-L1@EGO-900 formed core-shell structure while Fe-L3@EGO-900 formed spherical nanoparticles. This is unique and completely different from the known literature where similar thermal treatment produced non-core shell structure. The reason can be ascribed to the type of carbon support used in our work and its role in the formation of the nanocatalyst. The activity of the nanocatalysts in the dehydrogenation reaction clearly showed Fe-L1@EGO-900 gave better activity than pure phases or support pointing out the role of interfaces of mixed phases of iron.

These discussions are included in the revised manuscript.

What exactly is the difference between the catalyst obtained by treatment at 900 degrees C and those at other temperatures?

Response: PXRD analysis of Fe-L1 on exfoliated graphene oxide (EGO) treated at different pyrolysis temperatures such as 400 °C, 600 °C, and 900 °C is shown in the Figure S7. A broad peak ranging between $2\theta = 20 - 30$ degrees was observed for Fe-L1@EGO-400. However, this region has characteristics of graphitic peak and an unidentified peak. The feature retained even at 600 °C pyrolyzed sample. The reason could be due to undecomposed ligands on EGO that formed poorly crystalline carbonaceous materials. However, Fe-L1@EGO-900 that was prepared by pyrolyzing at 900 °C, a clear sharp peak corresponding to graphitic basal plane emerged at $2\theta = 26.5^\circ$. Additionally, pyrolysis at higher temperatures also resulted in the formation of crystalline phases of oxides, nitrides, and carbides. From PXRD data (Figure S7) and TGA (Fig. 8), it is clear that at high pyrolysis temperature (~ 900 °C) the iron complex decomposed to form iron oxide, nitride, and carbide phases.

Figure S7. (i) PXRD of Fe-L1@EGO-400, (ii) Fe-L1@EGO-600, (iii) Fe-L1@EGO-900 catalyst.

The effect of pyrolysis temperature on the distribution of size and morphology of the nanoparticles were studied using TEM analysis (Figure S9). It is clear from the Figure S9 a, b that the catalyst treated at 400 °C is forming well distributed small iron nanoparticles in the size range of 2-10 nm. The individual nanoparticles were not in specific shape and there is no core-shell type of morphology as shown in the panel b. As the temperature was increased from 400 °C to 600 °C, a few particles were agglomerated and form bigger particles with an average size of 10-12 nm as shown in panels c and d. In the similar way when the temperature was increased to 900 °C, iron-rich crystalline nanoparticles were well dispersed on the carbon support with size ranging between 5-100 nm (most of the nanoparticles are having size 12-15 nm shown; see in panels e & f). The individual nanoparticles with specific core-shell type morphology could be

observed only at higher magnification. It is clear from the PXRD and TEM images that treatment temperature is clearly affecting the composition as well as distribution, size, and morphology of the individual particle. Thus, the composition as well as the nature of nanoparticles is affecting the catalytic activity. This discussion is included in the revised supporting information.

Figure S9. **a & b)** TEM images of Fe-L1@EGO-400 at the scale bar of 50 nm and scale bar of 20 nm; **c & d)** TEM images Fe-L1@EGO-600 at the scale bar of 50 nm and scale bar of 20 nm; **e, f, g & h)** TEM images of Fe-L1@EGO-900 at the scale bar of 50 nm, scale bar of 20 nm and scale bar of 5 nm.

In the "discussion" the authors refer to an "interesting correlation of the microstructure and catalytic activity". Undoubtedly remarks can be found buried somewhere in the "Results" section but there needs to be a discussion in the "discussion" section.

Response: We have addressed this point in the revised manuscript with an adequate data.

The manuscript also contains several typos which need to be removed, e.g. expoliation vs exfoliation.

Response: Thanks. Typos are corrected.

Reviewer #2 (Remarks to the Author):

The authors improved their manuscript as suggested by the reviewers. With the new citations added, T. Schwob, R. et al. *Angew. Chem. Int. Ed.*, 2016, 55, 15175-15179 should be included as well. I suggest publication after this minor issue has been addressed.

Reviewer #2: The authors improved their manuscript as suggested by the reviewers. With the new citations added, T. Schwob, R. et al. *Angew. Chem. Int. Ed.*, 2016, 55, 15175-15179 should be included as well. I suggest publication after this minor issue has been addressed.

Reply: We thank this reviewer for accepting our article for publication. The suggested reference is included in the revised manuscript (Ref.55).